# Developing the Next Generation of Augmented Reality Games for Pediatric Healthcare: An Open-Source Collaborative Framework Based on ARCore for Implementing Teaching, Training and Monitoring Applications

**DOI:** 10.3390/s21051865

**Published:** 2021-03-07

**Authors:** Aida Vidal-Balea, Óscar Blanco-Novoa, Paula Fraga-Lamas, Tiago M. Fernández-Caramés

**Affiliations:** 1Department of Computer Engineering, Faculty of Computer Science, Universidade da Coruña, 15071 A Coruña, Spain; aida.vidal@udc.es (A.V.-B.); o.blanco@udc.es (Ó.B.-N.); 2Centro de Investigación CITIC, Universidade da Coruña, 15071 A Coruña, Spain

**Keywords:** augmented reality, mixed reality, gaming, ARCore, teaching, training, online education, pediatric, mobile health, eHealth

## Abstract

Augmented Reality (AR) provides an alternative to the traditional forms of interaction between humans and machines, and facilitates the access to certain technologies to groups of people with special needs like children. For instance, in pediatric healthcare, it is important to help children to feel comfortable during medical procedures and tests that may be performed on them. To tackle such an issue with the help of AR-based solutions, this article presents the design, implementation and evaluation of a novel open-source collaborative framework that enables to develop teaching, training, and monitoring pediatric healthcare applications. Specifically, such a framework allows for building collaborative applications and shared experiences for AR devices, providing functionalities for connecting with other AR devices and enabling real-time visualization and simultaneous interaction with virtual objects. Since all the communications involved in AR interactions are handled by AR devices, the proposed collaborative framework is able to operate autonomously through a Local Area Network (LAN), thus requiring no cloud or external servers. In order to demonstrate the potential of the proposed framework, a practical use case application is presented. Such an application has been designed to motivate pediatric patients and to encourage them to increase their physical activity through AR games. The presented games do not require any previous configuration, as they use ARCore automatic surface detection technology. Moreover, the AR mobile gaming framework allows multiple players to engage in the same AR experience, so children can interact and collaborate among them sharing the same AR content. In addition, the proposed AR system provides a remote web application that is able to collect and to visualize data on patient use, aiming to provide healthcare professionals with qualified data about the mobility and mood of their patients through an intuitive and user-friendly web tool. Finally, to determine the performance of the proposed AR system, this article presents its evaluation in terms of latency and processing time. The results show that both times are low enough to provide a good user experience.

## 1. Introduction

Augmented Reality (AR) has been drawing more and more interest in the last years both for industrial and entertainment purposes [1,2]. There are an increasing number of AR solutions, but very few of them consider fast reaction times or provide shared experiences. Such features enhance dramatically human-to-machine interaction, since they enable building real-time interactive systems for fields like industrial automation, healthcare or gaming. In addition, shared experiences add the possibility to immerse multiple users in the same AR scenario in a way that they can interact simultaneously and with the same virtual elements.

Specifically, the term AR refers to a technology that provides an environment in which virtual objects are combined with reality, thus integrating computer-generated objects in the real world [3]. Therefore, AR combines real and virtual objects in a real environment, runs interactive applications in real time, and aligns real and virtual objects with each other [4]. When AR is combined with the ability to interact with the real world through virtual objects, the term Mixed Reality (MR) is frequently used. Due to such capabilities, latency is a key factor in both AR and MR shared experiences, since it impacts user experience by desynchronizing the visualization and reactions of virtual elements, which is essential in a wide variety of fields, such as in healthcare (e.g., in therapy or rehabilitation processes [5]), teaching [6] or in industrial environments [7].

This article presents a novel collaborative framework that is able to synchronize shared experiences where multiple AR devices detect each other and interact through a Local Area Network (LAN) without depending on a remote server. Thus, the framework allows for deploying shared AR applications anywhere, even without an Internet connection, as long as the devices are connected to the same local network.

In addition, this article proposes a practical use case where an AR mobile application is presented, describing the design and development of an innovative architecture for AR gaming experiences. Such an application is focused on helping pediatric patients and includes the necessary tools and accessories to monitor its use and the physical condition of the patients according to their application usage patterns. Since the proposed type of application should facilitate and motivate the physical activity of pediatric patients, the use of AR is essential, because it provides the ability to use the real world environment as a playground. In this way, it is possible to create three-dimensional activities that foster player physical activity in order to interact with the game. Moreover, the AR mobile gaming framework described in this article allows multiple players to engage in the same AR experience, so children can interact and collaborate among them sharing the same AR content, which is placed at the same positions.

Furthermore, the proposed system includes a web data crowdsourcing platform for doctors and medical staff that is able to manage pediatric patient profiles and that facilitates the visualization of the data collected by the mobile application. Such a web tool also allows for configuring the different parameters of the AR system so as to restrict access or to enable the specific activities that each children can perform with the mobile application. Thus, the developed system provides a very useful tool for developing AR experiences focused on patients with long pediatric stays at the hospitals.

Specifically, this article includes three main contributions:It provides a detailed description on how to develop a novel open-source collaborative AR framework aimed at implementing teaching, training and monitoring pediatric healthcare applications.It details how to make use of the developed AR framework so as to implement from scratch a practical pediatric healthcare application. Such an application does not need any previous configuration from the user and includes the possibility of creating shared AR experiences between different mobile devices. In addition, the developed system enables the collection and visualization of custom usage data, which can be remotely monitored and analyzed.The performance of the framework is evaluated in terms of latency and processing time in order to demonstrate that it provides a good user experience. In addition, it is worth noting that the presented framework is open-source (under GPL-3.0 license), so it can be downloaded from GitHub [8] and then be used and/or modified by researchers and developers, who can also replicate the experiments and validate the provided results.

The rest of this paper is structured as follows. Section 2 reviews the state of the art on augmented and mixed reality collaborative applications, and analyzes some the most relevant academic and commercial mobile gaming solutions for pediatric patients. Section 3 details the design and implementation of the proposed collaborative AR framework. Section 4 illustrates with practical uses cases how the proposed collaborative AR framework can be used for developing applications for pediatric patients. Finally, Section 5 details the experiments carried out to evaluate the framework performance, while Section 6 presents the conclusions.

## 2. State of the Art

### 2.1. Collaborative Augmented and Mixed Reality Applications

AR and MR are currently considered two of the most promising technologies for providing interfaces to visualize and explore information [9] and they present an opportunity to redefine the way in which people collaborate in fields like telemedicine [10] or intelligent transportation [11].

In the literature there are some recent preliminary works of collaborative AR/MR applications. The most promising developments use expensive smart glasses like Microsoft HoloLens [12]. For example, Chusetthagarn et al. [13] presented a preliminary Proof-of-Concept (PoC) for visualizing sensor data in disaster management applications. Such a work makes use of HoloLens spatial anchors through a built-in sharing prefab provided by Holotoolkits, a Unity package to create a collaborative AR/MR environment. Unfortunately, such an implementation is currently considered as deprecated. It is also worth mentioning that Microsoft has been working on a Microsoft HoloLens sharing framework, which, among other features, includes a discovery process through UDP. However, although the mentioned Microsoft’s development was probably the most promising collaborative framework solution, it has been recently notified that the framework will be no longer maintained [14].

There are other recent works devoted to providing mobile collaborative AR experiences. For example, Zhang et al. [15] proposed a client–server based collaborative AR system that integrates a map-recovery and fusion method with a vision-inertial Simultaneous Localization and Mapping (SLAM). The authors validate the precision and completeness of their methods through a number of experiments. In addition, the launch of Google’s ARCore [16] and Apple’s ARKit [17] have simplified substantially the development of mobile AR applications [18]. However, no collaborative applications based on such software have been found in the literature, but, with the prospect of 5G/6G networks, it has been an uprise in the number of web-based mobile AR implementations that rely on efficient communication planning [19].

The concept of collaboration can be also understood as a way of enabling the interaction between Internet of Things (IoT) and AR/MR devices [20]. For instance, in [21] the authors presented a PoC of metal shelving that is monitored with strain gauges and that has a QR code attached. When the operator scans the QR code, certain identification data are sent to a cloud and a simulation model designed with Matlab provides a stress analysis that is visualized through a pair of F4 smart glasses. Other authors focused on enabling automatic discovery and relational localization to build contextual information on sensor data [22]. In the case of the work detailed in [23], the researchers describe a scalable AR framework that acts as an extension to the deployed IoT infrastructure. In such a system, recognition and tracking information is distributed over and communicated by the objects themselves. The tracking method can be chosen depending on the context and is detected automatically by the IoT infrastructure. The target objects are filtered by their proximity to the user. It is also worth mentioning the work of Lee et al. [24], which proposed an architecture to integrate HoloLens through a RESTful API with Mobius, an open-source OneM2M IoT platform. However, in such a work the authors considered that further work will be needed in order to consider the various requirements defined by OneM2M.

### 2.2. Mobile Gaming for Pediatric Patients

#### 2.2.1. Academic Developments

A number of articles endorse the use of technologies like AR or Virtual Reality (VR) as a mean through which the quality of life of pediatric patients can be improved. For example, Gómez et al. [25] proposed the use of VR in order to alleviate the negative side effects of chronic diseases that lead to periodic hospital admissions. Some of these symptoms can be anxiety, fatigue, pain, boredom or even depression, among others. Corrêa et al. [26] presented an AR application for conducting activities related to musical therapy, which can be utilized on motor physical therapy. Burdea et al. [27] and Pyk et al. [28] used VR and video games for upper-limb physical therapy. For such a purpose, the authors of [27] made use of a modified PlayStation 3 game console, 5DT sensory gloves and other hardware specifically designed for children. Martínez-García et al. [29] developed “PainAPPle”, a mobile application that allows for measuring pain levels on children. This latter initiative tries to overcome the difficulties that hospital staff face when assessing the pain levels on young children that cannot talk or patients that have problems for expressing their feelings. Thus, in the application proposed by the authors, children can use images and shapes to express their mood.

#### 2.2.2. Commercial Applications

Different commercial applications are available for easing pediatric stays. For instance, there are projects such as “Nixi for children” or *“Me van a hacer un transplante”* (“I’m going to have a transplant”) whose main objective is to inform and reassure pediatric patients before operations in order to reduce their fear and uncertainty. In the case of “Nixi for children” [30], the aim of the project is to reduce the pre-operative anxiety on children. For this purpose, children use a VR device through which they can see immersive 360 videos where the procedures on the operation are explained so that pediatric patients become familiar with the environment. Regarding *“Me van a hacer un transplante”* [31], it is an application where it is explained to children what a bone marrow transplant is and what steps it consists of, in a way that makes it easier for them to understand the procedure they will undergo. This application includes a tale, a video and three games that children can play with while learning.

There are other applications that were devised with the idea of facilitating the communication between children and health-care personnel through different games and activities. For example, “Dupi’s magic room” [32] is a project aimed at developing a mobile application with which children can express themselves and explain their feelings through interactive games and drawings. Thus, healthcare professionals can monitor patient mood over time and check their progression. This application utilizes psychopedagogical techniques that allow for analyzing drawings in order to determine the patient emotional status, therefore helping children to express their feelings effortlessly.

Regarding patient stress management, there are a few applications aimed at reducing pre-operative or post-operative anxiety, although they are mostly focused on adult patients. An example is *“En calma en el quirófano”* [33] (’At ease in the operating room’), which is designed to guide patients through sounds so that the user is able to cope with the preparation prior to entering the operating room or to performing a medical test. Similarly, the application *“REM volver a casa”* [34] (REM go home) provides the user with videos and sounds that allow the patient to follow a mindfulness training program.

Finally, it is worth mentioning two applications designed specifically to entertain children during long hospital stays: “RH Kids” [35] and “EntamAR” [36]. Regarding “RH Kids”, it provides educational material for children. It includes different interactive stories, educational games and working sheets in order to have a positive impact on children happiness, which leads to a more efficient treatment and to shorten the time at hospital. “EntamAR” is an application that is similar to the one described in this article, since it is an AR mobile video game that aims to improve the quality of life of pediatric patients. Such an application makes use of the Onirix AR platform [37], through which the rooms can be scanned and used afterwards to build games. Thus, both the application and each of the activities included in it have to be developed ad hoc for each hospital, which requires that a technical team (usually volunteers or healthcare personnel), invest a relevant amount of time and effort in the design and implementation of each application. In addition, if the physical space in which the application is executed changes significantly, the room would have to be re-scanned and each of the activities reassembled.

### 2.3. Analysis of the State of the Art

After reviewing the state of the art, it can be concluded that there are recent works with the aim of providing AR/MR shared experiences. However, in contrast to the open-source collaborative framework presented in this article, the vast majority of previous works are very early developments with a relevant number of open issues that require further research. In addition, most of them rely on advanced network and communication capabilities (e.g., 5G), remote servers, or sophisticated expensive AR/MR devices.

Regarding pediatric healthcare use cases, all the solutions cited in Section 2.2.1 and Section 2.2.2 either solve totally or partially issues related to long pediatric stays. Such solutions can be classified into three categories: alternatives that facilitate the life of patients in health environments regardless of their age, solutions that help children outside of hospital environments (but that can be adapted to be used in such environments) and applications that currently exist to entertain children during long-term hospital stays.

Among the analyzed applications, only a few target children and there is only one solution that makes use of AR. Regarding this aspect, “EntamAR” uses a platform which needs an ad hoc solution for each game and location where the game is to be played. This fact involves the need for carrying out a previous process in which the room has to be scanned and the game has to be re-assembled using the mentioned AR platform.

Therefore, after analyzing the previously mentioned alternatives, it was concluded that there is a lack of autonomy of the AR systems when it comes to provide a complete User Experience (UX). Moreover, none of the analyzed alternatives offer a mechanism for monitoring patients in order to visualize their state or evolution.

Due to the previous issues, the framework presented in this article is focused on enabling the development of AR applications that motivate the mobility of hospital patients and that needs no previous configuration from the user. In addition, the open-source framework provides a novel feature that has not been found in the state of the art: it allows pediatric patients to share the same AR experience in real time, so they can collaborate among them when playing the games. Furthermore, the developed system enables the collection and visualization of patient usage data. Such data can be really useful for doctors, who will have a dynamic and indirect way for collecting data on the mobility and mood of their patients.

## 3. AR collaborative Framework

The next subsections describe the internal components of the developed framework. Specifically, it is first detailed the communications architecture used by the AR applications that make use of the collaborative framework and, next, the design and implementation of the framework are described thoroughly.

### 3.1. Communications Architecture

Figure 1 shows an overview of the architecture of the developed framework, which is divided into three parts: the visualization subsystem, the mobile application and the backend server.

The backend hosts a remote database and manages the requests made from web and mobile applications through a Representational State Transfer (REST) Application Programming Interface (API). In this article it is assumed that a mobile application (which is assumed to be implemented as an Android app, since such an operating system is used by nearly 80% of current mobile devices) is in charge of gathering and managing the data collected from the users and is responsible for storing such data on a local database on each device. Moreover, the mobile application is responsible for sending usage data to the backend server.

As far as data storage is concerned, only the minimum necessary information is stored in the mobile device local database in order to guarantee the proper functioning of the application. For instance, in a pediatric healthcare application, the remote database would store all the essential personal information on the patients: height, the difficulty level of the games, year of birth and the dates when the users were created and modified.

All the AR mobile devices within the same LAN connect with each other without requiring to connect to the Internet. Such a communication is handled by the framework, which is able to keep track of the status of each user and propagate the events that occur in each of the application instances.

### 3.2. Design and Technology Selection

During the design of the proposed collaborative framework, the following requirements were considered:The devised system has to support AR visualization and interaction.It has to allow two or more players to place 3D objects in a way that they are synchronized regarding materials or animations performed on the objects.If new objects are incorporated into the scene, they have to appear on all of the connected mobile devices at the appropriate positions.The system has to provide scoreboards and a method to keep them synchronized at all times.The system should allow an easy method to create, host and join games, keeping in mind that the application that would use the proposed framework could be used by children.The system has to provide a way to store usage data from users and gather them in a centralized location, so it can be easily accessed and analyzed.

Considering the previous requirements, the proposed framework was designed to provide a set of tools to develop AR games for pediatric healthcare that is composed by the following components, which are described in the next subsections: the AR subsystem, the backend server, the communications subsystem and the visualization subsystem.

#### 3.2.1. AR Subsystem

The AR subsystem is divided into the following sub-components:Augmented Reality and game engine. It must be first noted that there are actually two main types of AR: marker-based and markerless AR [2]. In addition, AR content can be displayed through different devices, being the most common mobile phones and tablets. However, in the last years Head Mounted Display (HMD) devices have evolved significantly, providing new visualization systems for AR and adding the possibility of having stereoscopic vision. Considering the requirements of most pediatric healthcare scenarios (similar to the ones indicated later in Section 4.2), it was concluded that the best alternative for the proposed application consisted in making use of markerless AR, since the other types of AR require the use of external infrastructure (i.e., markers) or are not suitable for the application target user (i.e., smart glasses). Among the different available AR platforms, ARCore [16] was selected for the development of the AR activities, since it specializes on surface detection, it has official extensions and packages for the most popular game engines and it offers an increasing number of compatible devices. Regarding the AR development tool, the game engine options are quite restricted, since they have to provide ARCore support. The options offered by the official ARCore website are Android [38], iOS [39], Unity [40] and Unreal Engine [41]. After considering the two available graphic engine alternatives, it was decided to use Unity due to its large community, which provides numerous examples, tutorials and forums. In addition, the learning curve of Unity is less steep than the one of other tools like Unreal Engine. Specifically, Unity 2019.3 was selected, since it is currently the only version that provides a direct way to export a Unity project as an Android library. This functionality significantly reduces the complexity of the integration process of the AR games with the rest of the components of the mobile application.Mobile Platform. For the selection of the mobile platform, the considered alternatives were the two most used platforms at the moment (Android and iOS), as well as the use of a hybrid mobile development framework. After considering the compatibility with the libraries provided by ARCore and with the used game engine, it was decided to develop the application for Android, since, as it was previously mentioned, it is currently the most popular mobile operating system, it has a wider support and a growing number of ARCore-compatible devices. However, other platforms might be taken into consideration in future work.Local database. The local data storage system is located on each mobile device and stores all the data collected by the application. For each patient, information is collected about his/her daily mood survey results, as well as the playing time and his/her step count (to quantify the amount of performed physical exercise). In addition, the patient profile settings are stored on the local database. All the mentioned data and settings can be also visualized and modified from the web application. In the case of Android, local storage can be implemented through the shared preferences, the internal/external storage system or by using a local database [42]. Due to the nature of the stored data, which are confidential, a data storage system with restricted access is needed. In addition, it was considered that the data to be stored are structured and the fact that the storage system will be read and written with medium to high frequency. For such reasons, it was decided that the option that best suits the requirements of the framework is a local database like Room [43].

#### 3.2.2. Backend Server

The backend server was designed as a web server that follows the Model View Controller (MVC) pattern. Such a software architecture pattern separates the data presentation layer from the management logic of the user interactions. The server is therefore in charge of managing the business logic and the data access layer, including all the infrastructure that supports the provision of the provided services.

The backend server provides a REST API through which various endpoints can be accessed.

Server software. Different technologies (frameworks and programming languages) can be used for implementing the backend server, like Java (e.g., Spring Boot), C# (e.g., .NET), JavaScript (e.g., Node.js) or Python (e.g., Django). Among the available solutions, Django was selected due to its learning curve, available documentation, libraries and extensions (e.g., chart generators, REST APIs, serialization libraries or database access libraries through Object-relational Mapping (ORMs)), community support, security and scalability.Remote database. It can be implemented by using cloud storage (e.g., Amazon Web Service (AWS)) [44], Azure [45] or Google Cloud [46]) or through a local database (e.g., MySQL [47], PostgreSQL [48] or MongoDB [49]). After considering the different alternatives, it was decided to make use of a local database, mainly to avoid storing data out of the hospital/healthcare network. There are different databases officially supported by Django (PostgreSQL, MariaDB, MySQL, Oracle and SQLite) [50], among which SQLite was chosen due to its very simple configuration and its low computational resource consumption.

#### 3.2.3. Visualization Subsystem

Different technologies can be used for developing the Visualization Subsystem, like Angular [51], React [52], Vue [53], Bootstrap [54] or Materialize [55]. A strict requirement is that the app should be able to make asynchronous requests to the REST API of the backend. In addition, other aspects were considered (e.g., learning curve, responsive design, implementation complexity, available documentation) and, eventually, Bootstrap was selected.

#### 3.2.4. Communications Subsystem

The development of shared AR experiences requires the use of a system that facilitates the communications among AR devices. For the proposed system, the mentioned system has to be compatible with Unity and should provide an easy way to implement the requirements of the AR games: to spawn virtual objects at a specific spot, to detect which user has first performed certain action and to keep the scoreboards synchronized.

The first alternative to develop the framework may consist in using the tools provided by Unity. In this case, it would be necessary to use the UNet network library [56], which provides a wide range of functionalities that are needed to develop online games. The main drawback is that UNet is deprecated since 2018 and will be removed from Unity, so its future software support will be a problem.

Another alternative provided by Unity is Multiplay [57], a game server that hosts services based on a consumption model, which implies that users have to pay for it as much as they use it [58]. In addition, Multiplay was still in alpha when the development of the work presented in this paper started, so it was decided to look for other alternatives.

The other two best known options are Mirror [59] and Photon [60]. Both are quite similar, as they provide basic networking capabilities, as well as client-to-client connections via a server and connections where one of the clients hosts the game by acting as the server. One advantage that Photon has over Mirror is the matchmaking service, although this does not make a big difference respect to Mirror when implementing two-player games, as this drawback can be easily solved through software. Moreover, it must be noted that Mirror and Photon differ in their cost, since Photon charges for its services and Mirror is completely free.

Considering the previous analysis, it was decided to use Mirror as the baseline for the implementation of the collaborative framework. This is due to its similarity with the deprecated UNet and the large number of users and documentation they offer. In addition, the fact that Mirror is a free software project means that the code can be accessed at all times, and can even be easily modified or extended if necessary.

Figure 2 shows a more detailed view of the designed framework. Two players are illustrated on the diagram: the one on the left acts as a local server, providing connectivity for the rest of the users and working as a coordinator; the other player only needs to join the game and the application will be executed as a client. This approach provides the chance to make the game portable, as it is a serverless application, hence it does not depend on an external server.

## 4. Practical Use Case

### 4.1. Description of the Use Case

One of the priorities that is always present in a hospital environment is to improve the quality of life of patients. For instance, one of the consequences of hospitalization is an increase in patient stress levels, including children, who usually experience increases in their stress levels, which derive into aggressive behaviors, isolation and difficulties when recovering from medical procedures [61]. Minimizing anxiety is important in order to help children to be more relaxed and comfortable during interventions and medical tests that may be conducted on them while they are at the hospital. Children with less anxiety will cooperate more easily, will be less afraid of procedures, will need less sedatives and will have shorter recovery times [61,62].

During periods of hospitalization, play activities are often used in the form of therapeutic play or play therapy, thus providing an improvement in their physical and emotional well-being, while also reducing patient recovery time. Games can help to reduce the intensity of the negative feelings that occur during children’s hospitalizations, thus being appropriate, for example, in pre-operative period preparations and invasive procedures [63]. Therefore, the use of games inside a hospital can become an interesting tool to be considered for the assistance to hospitalized children. The use of AR also increases children engagement and motivation, preventing them from getting bored and encouraging them to use the application for longer periods of time.

To show the technical capabilities and the performance of the proposed AR framework, a demonstrative experience was created. In such an application, part of the games were adapted so that they can be used both by children with complete or restricted mobility. It is important to note that the application is intended to test the developed system from a technical point of view and demonstrate its technical capabilities. No empirical scientific tests with children were made. Any applications made with this framework will have to be assessed taking into account medical considerations and tests will have to be carried out. In our future work we will tackle these issues as soon as the COVID-19 pandemic allows for carrying out the experiments in safe scenarios. For example, one possibility would be to perform the evaluation as follows. First, the testing group would be split into two subgroups: one that would use the application daily and another that would not. The idea would be to conduct periodic surveys during their hospital stay asking for changes on, for instance, children’s behavior, pain level or mood improvement, among others. The final stage of the experiment would consist in comparing the results of the two groups and verify the effectiveness of the proposed application. Another useful experiment would require to conduct surveys before and after children undergo medical tests, thus evaluating and comparing the results to verify if the mobile application is also effective in such scenarios. At the same time, throughout this evaluation process, the usability of the application would also be examined, looking at how children use the application and including questions in the surveys about such a topic.

### 4.2. Main Requirements of the System

Considering the system objectives mentioned in Section 4.1, the following are its main design requirements:AR should be included in the mobile application in order to encourage children to perform daily physical activity. Since AR can integrate the virtual elements of the game into a real scenario, it is possible to provide new augmented experiences and to increase player engagement and motivation.Children are the target of the AR application, so the User Interface (UI) needs to be as simple as possible and the mobile device should be comfortable and easy to use. For these reasons, the use of mobile devices such as mobile phones or tablets are a better option than smart glasses or HMD devices. In addition, HMD devices like Microsoft HoloLens smart glasses are much more expensive, they are designed for adult-size heads, they are quite heavy (579 g) and have a steep learning curve.The mobile AR application should be designed to host games aimed at children between 6 and 14 years old. This decision implies that the UX design needs to be adapted to the mentioned age range. As a consequence, the AR application has to implement simple interfaces, large buttons and actions that require a minimum number of steps.In order to be able to run the application on different mobile devices, its computational resource requirements need to be limited, thus emphasizing its performance rather than visual details. For such a reason, cartoon low-poly aesthetics are appropriate, since they help to reduce computational complexity.The mobile AR application needs to collect the children’s usage data so that they can be later visualized and managed through a web platform as well as through the developed mobile application. Such data should be displayed by using simple charts, where playing time, steps and daily surveys can be shown in an attractive way.It is important to consider that the games to be designed are intended to encourage patients to engage them in moderate physical activity inside a hospital within a restricted area. For example, as it will be described later in Section 4.7.1, in one of the games, which is inspired by the well-known “Marco Polo” game, a player is hidden and the others have to find the person using their voice as a guide. In this case, the hidden player would be replaced by the mobile device, which would emit different sounds to indicate its position. Another example to encourage patients to walk and move around a room is the game detailed in Section 4.7.2, which creates 3D animals as the user walks and interacts with them.The proposed games should also be multi-player games and ideally foster the cooperation of the pediatric patient with other people (e.g., parents, medical staff or other patients) in order to solve a problem. For instance, the game described in Section 4.7.3 involves two people, who play different roles: one player sees a 3D map, while the other one has the clues that are needed to solve a puzzle. Thus, the two players have to communicate and cooperate in order to accomplish the indicated goals. In addition, another game has also been developed, described in Section 4.7.4, in which players compete to see which of the two obtains more points by capturing virtual animals that are displayed on the physical scenario.The requirements for playing the game should be minimized. Therefore, the use of external components (e.g., QR markers, sensors) or previous scenario-recognition phases should not be necessary so as to avoid scanning the environment and thus removing the need for creating and personalizing the games for each hospital room. As a consequence, the developed solution should be flexible enough to be used during the most common situations and should be adapted to the abilities and the most common limitations of a pediatric patient.

### 4.3. Mobile AR Application

Following the requirements previously indicated in Section 4.2, the mobile AR application was designed having in mind that it is going to be used by children (by pediatric patients aged between 6 and 14), so the app interface and interactions with the users have been devised to be as simple as possible, with visual cues and minimizing the amount of text to be read.

Figure 3 shows a diagram that illustrates the relationships among the components of the application, as well as the interaction flow that the user follows depending on different parameters. As it is illustrated in Figure 3, when the application is opened, the patient has the option to login by scanning a barcode (which is one of the most common identification technologies used for identifying patients in a hospital) or he/she can skip this step if identification is not needed. If a barcode is not scanned, the user menu is displayed and he/she can choose from the different available AR games.

In case the user logs in by scanning a barcode but does not grant the data storage permission, the application will not collect or store information about the patient. In contrast, when the user logs in by scanning a barcode and accepts the necessary storage permissions, it is checked whether the daily survey is required (if it is required, the survey will be displayed). When the user completes the survey or in the case that it is not necessary to do so, the user menu is shown, which enables accessing the AR games and viewing the user statistics stored in the local database of the device.

The data collected by the mobile application are stored so that they can be viewed by the medical staff and patient families in a quick and efficient way. The data collection process consists of two steps: first, the mobile application collects and stores data in its local database; then, if an Internet connection is available, the data are sent to a server, where they are stored permanently.

The devised application design also considers other usability aspects in order to adapt the controls and interfaces to the different constraints of the mobile application users. For instance, the interface is as simple as possible, with a limited amount of different screens, so users do not need to navigate through multiple screens to get to the different parts of the application. In addition, the amount of shown text has been minimized and icons have been incorporated, so that children can easily understand how to use the application.

Regarding the four games described later in Section 4.7.1–Section 4.7.4, they were designed in a different way, having in mind the different situations that may arise in a hospital, where children suffer from different conditions that hinder their mobility. For instance, the mentioned mobility constraints led to allow users to move the maps of the games wherever they want and to let users rotate and scale such maps, thus facilitating accessibility to all parts of the map for those patients who cannot move easily.

Finally, attention should be paid to the fact that environmental conditions affect ARCore performance when detecting and tracking surfaces [64]. For this reason, during the development of the games, the environmental factor has been considered while testing them. For such a purpose, tests have been carried out in different lighting conditions and different surfaces: tiles, wood, monochromatic lacquered tables and concrete. In addition, the speed of movement and the rotations that the user can make while using the device was evaluated. The application was tested under similar conditions to those that might be encountered in hospitals (i.e., inside classrooms, hallways, living rooms and work environments) and it has been concluded that the application works smoothly. Nonetheless, it was confirmed that the worse the light or the more homogeneous the surface is, the longer ARCore takes to detect the surface and the worse the tracking continuity. These drawbacks are mitigated to some extent thanks to the ability of ARCore to recover from these failures, as it is able to restore previous virtual object positions once the environment is recognized again.

### 4.4. Backend

In the proposed use case, the backend is used to save the data collected from the mobile devices, providing an information access point to the healthcare personnel and to patient’s family. This means that healthcare professionals do not have to collect manually the local usage data from each device. In addition, having a central database allows the information to be synchronized among different devices. Furthermore, the backend database can be used to back up local data, allowing for installing and uninstalling the application without losing patient data.

Three GET requests were designed: one for obtaining user profile information; another for determining whether the patient has already filled his/her daily survey; and a last request, devised for the website, which provides the summary of all the patient data so that they can be easily displayed on charts. In addition, the REST API defines three POST requests that are used to send information to the server regarding new users, games and surveys.

### 4.5. Web Frontend

The web frontend is the interface through which remote users and the backend server communicate. Since the backend server was designed by using the MVC pattern, the frontend corresponds to the view part of the pattern.

To implement the frontend, a website was devised to visualize and manage patient data remotely. Specifically, the website was designed to provide easy navigation and access to each patient’s data in a clear and simple way. To access each patient’s data, the remote viewer only has to enter the barcode number of the monitored patient. The website also provides a menu to modify the patient settings and restrictions of the games.

Specifically, the web application offers the following pages that users can navigate through:Home page. It provides a welcome page with a video showing all the games and a summary on the features of the mobile and web applications. In addition, a button is provided through which the remote user can access the patient search page. A screenshot of the home page is shown in Figure 4 on the left.Search page. It is used to look for a patient profile by inserting his/her barcode number. Figure 4, in the center image, shows a screenshot of the search page.User data edition page. In this page the patient’s personal information can be edited. A screenshot of the edition page can be seen on Figure 4 on the right.Patient data page. It can be accessed from the search page after registering a patient in the database. As it can be observed in Figure 5, the patient data page contains the patient personal information together with charts that show his/her step count and the evolution of the answers of patient daily mood survey.

In order to validate the proposed mobile AR framework, the use cases described in the next subsections were tested, verifying the authentication and initialization of the system, and each of the developed games, as well as the web application.

### 4.6. Authentication and System Initialization

Within the mobile application, there are two available options: the user can access with or without scanning the barcode. In order to use the application and store the patient’s usage data, the patient must first log in. The way to log into the application is by scanning the barcode of the hospital identification bracelets, as it can be seen in Figure 6.

Once the barcode is scanned, it is checked whether the patient is registered in the system. If the patient is not registered, such a registration is then performed by asking for permission to collect and store his/her data. Then, the login process continues by checking whether the patient must take his/her daily survey and, then, the main menu is displayed. During this process, if an Internet connection is available, requests will also be made to the server to check if there are any data about the user on the backend server.

The daily survey is shown during the login process once a day for each registered patient. As it is shown on Figure 7, this survey collects data on a scale of 1 to 5 on mood, pain level and appetite.

If a user does not want to register or send his/her usage data, he/she would choose the login without barcode option, in which it is only possible to access the application and start a gaming session.

### 4.7. Designed Games

Within the Unity project there is a main scene where the rest of the scenes are loaded. This scene has an object to which a parameter is passed with the scene number that has to be loaded next.

ARCore is used to detect horizontal surfaces, which is necessary for the correct positioning of 3D objects on real scenarios. During this process, raycasting is performed from the camera in the direction of the user to check for collisions with the surfaces detected by ARCore. The collision point is where the virtual objects will be placed.

The system of user interfaces, that includes the buttons and the interactions with the user through scripts that can be adapted to each specific scene, is the same for all scenes. Users can show/hide the main menu and restart the game. In addition, in the “Map explorer” and “WakaMole” games, users are able to scale and rotate the map in order to see it from a more comfortable point of view without having to move excessively. As it was previously mentioned, such a functionality is aimed at easing the interactions of patients whose reduced mobility prevents them from solving a challenge because they are not able to visualize the map from all the required perspectives.

Moreover, the developed mobile application enhances its UX through haptic feedback, which is used to reinforce the response of the system to user interactions. Thus, when the user makes a mistake, a short vibration is emitted, while when the game is completed a repetition of vibrations is emitted as a ‘celebration’. In addition, it is also considered appropriate to add a particle system that simulates confetti to indicate that an objective was achieved and to reinforce the positive feeling of the user.

Another aspect that is considered important is to adapt the size and position of the map scaling and rotation sliders when rotating the mobile device. In this way the shape of the sliders is not distorted, since Unity’s way of managing the interfaces is by ratios and distances to anchors. In the case of the screen rotation, it is an event that is collected by the Android application and then sent through a message system that a script receives in Unity.

#### 4.7.1. Marco Polo

The first of the activities devised for pediatric patients was the game “Marco polo”, which does not involve AR, but which enables testing the rest of the components of the developed system. Specifically, the game requires a minimum of two people: one would hide the mobile device and would click on a button, while the rest of the players will try to find the mobile device by being guided by the sound it plays.

In Figure 8, it can be observed the game main screen, where the user can select from one of the three difficulty levels that are available. Depending on the selected level, the sound, melody and its frequency are adapted. The greater the difficulty, the lower the sound and and its frequency.

Once the level is selected, the game can begin. A timer has also been added to keep track of the time from when the device is hidden to when it is found.

#### 4.7.2. Jungle Adventure

The main objective of this game is for the patient to walk and do a minimum of physical activity. The game consists of walking through a room or corridor, aiming the camera of the device towards the floor to detect the surfaces, where animals will appear. Figure 9 shows some examples of the animals while being automatically spawned on the floor.

When the animals are displayed, the patient will have to "collect" them by clicking on the 3D objects. When an animal is registered, its silhouette will light up on a panel located on the lower left side of the screen, making the user aware of the animals still to be found and showing how many of them have already been collected.

Once a surface is detected, an internal counter is triggered which, after a random time within a range, will cause an animal to appear on the surface that the user is targeting. The instantiated animal (3D model) will be randomly chosen from a list of animals that is passed to the scene controller for this purpose. After the user clicks on the animal, the counter will be activated again, restarting this cycle, which will stop when all the animals on the list are displayed.

#### 4.7.3. Map Explorer

This game challenges two patients to solve a map in a collaborative way. For this purpose, users can play two roles: the explorer, who has to guide a character through the map, and the assistant, who has the clues to discover the steps the explorer has to follow to solve the puzzle.

For this game there are two different maps available: Savannah and North Pole, which are shown on Figure 10. Each game consists in combining the clues given by the assistant player in order to solve the given challenge. These clues can be found on the mobile application, at the “MapAssistantActivity”screen, as it can be seen on Figure 11.

This activity uses the ARCore surface detection tool in order to place the map on a surface at the beginning of the game. In each map, the position of the characters and obstacles is predefined, changing in each execution small details that will be used to indicate the order of the resolution of the challenge. The correct order for the solution of each map can be found in the posts that can be accessed by the other player from their device. Thus, completing the game is a task that needs to be achieved with the collaboration between the two patients.

In addition, a timer has been included to count the time it takes the players to solve the challenge. This is meant to encourage the patients to keep playing and moving on a daily basis.

#### 4.7.4. Wakamole

This game was designed thinking about children who cannot move or have very little mobility, as they can play without the need for walking around a map. The game is for two patients that play against each other, trying to ’capture’ as many animals as possible. The animals appear periodically at random points on the map, and have a time limit after which they disappear. The player who first clicks on an animal gets a point. The game ends when one of the players reaches a previously agreed score, since there is no time limitation.

To perform the previously described process, the users need to be connected to the same game. For such a purpose, one of the players starts a new game, and the other one joins the game introducing a game code as it is illustrated in Figure 12, where the image at the center shows the host screen while waiting for the second player to enter the game code (which is illustrated on the image on the right).

After the matchmaking process is completed, the new scene containing the game is loaded. Figure 13 shows how ARCore is used for surface detection in order to place the map on a surface at the beginning of the game. When the game starts, the animals appear one by one for a fixed time. The first player that presses on the animal before it disappears gets a point. The spawning points of the animals are stored in a list, from which random points are taken each time a new animal is spawned.

If only one of the players clicks on the animal, there will be no conflicts, since there is enough time for the message to be sent from the client to the server (then the server will communicate the clients that the animal needs to be hidden, as it has already been picked by a user). The problem arises when both players click on the same animal almost simultaneously, as both event messages would reach the server and a point would be added to each player. In order to solve this problem, the message flow has been implemented in such a way that the client sends its own identifier to the server, and when the server notifies the clients that the animal has been clicked on, it also sends such an identifier. This allows each client to check if the received identifier matches its own before giving the order to add a point to its scoreboard. When one of the scoreboards is incremented, it is automatically updated in all clients, as they remain synchronized at all times. An example of the such scoreboards can be observed at the bottom of the screenshot on the right of Figure 13.

## 5. Experiments

The developed collaborative AR game described in Section 4.7.4 requires short response times to ensure a good user experience since, as it is a fast-reaction game, determining which user performs the actions first is critical to guarantee the fairness of the game and a smooth experience.

To evaluate the performance of the developed system, four sets of tests were carried out by using two different mobile devices. Such devices acted first as clients and then as hosts of the game in two different scenarios: through a local network (i.e., both users made use of the same WiFi) and through the Internet, when the two users were in remote locations. The used devices have the following specifications:Device 1 (tablet): Samsung Galaxy Tab S4 with 4 GB of RAM, 64 GB of internal memory and a Qualcomm Snapdragon 835 processor (Samsung Electronics Co., Ltd., Seoul, Korea).Device 2 (smartphone): OnePlus 6T with 8 GB of RAM, 128 GB of internal memory and a Snapdragon 845 processor (Oneplus, Shenzhen, Guangdong, China).

Each set of tests consisted in playing the game for 10 min. During such a time, the latency and the processing time data of every packet were collected and stored in a local file. It should be noted that the mentioned data were obtained in a way that they represent accurately the latency/time that the application will experience in a real environment. In addition, it is worth noting that, due to the single-thread nature of Unity, the measured times include the waiting times that the application requires during its execution in order to manage the processing slots of the used frames. Specifically, the following are the main steps included in the estimation of the latency:First, at one point during the game, a client sends a message to transmit a game event.Then, the server receives the message over the network.Next, the server waits until the next frame processing slot is ready.The received message is parsed and processed by the server, which sends a response to the client.The client receives the message from the server over the network.The client waits until the next frame processing slot is ready.The message is parsed and processed by the client.

Although the previous steps estimate times that are slightly higher than the existing network latencies (which are traditionally used by game servers to determine their performance), they provide more realistic and accurate estimations on the times that a user of the game will experience in real life.

Figure 14 shows a comparison of the obtained latencies for the two tested devices and for the previously indicated test scenarios. As it can be observed in the Figure, latency variability over time is not very high in the test networks. Such an observation is corroborated by the statistics shown in Table 1, which provides the mean, standard deviation and variance of the latencies plotted in Figure 14.

Both Table 1 and Figure 14 show that, as it could be expected, latency was slightly higher for the remote tests than for the local network. Nonetheless, it must be noted that, in other networks, the obtained values will vary depending on the characteristics of used network and on the number of connected devices, so the provided results should be considered merely as illustrative of a real use case.

Table 1 and Figure 14 also allow for determining the differences in the latencies experienced by the two evaluated devices: as it is indicated in Table 1, on average, the selected tablet is between 32 and 41 ms slower than the smartphone, which is essentially due to their different hardware.

Since, for each scenario, network latency is essentially the same (with slight oscillations due to traffic load), the observed latency differences are related to their processing time, which is the time required by the application to process a package and render the interface changes. Table 2 shows the processing times for both devices, which reach, on average, around 8 ms for the tablet and only 3 ms for the phone, thus corroborating the observed performance difference. As a consequence, future developers should be careful when choosing their AR devices, since, currently, they can impact user experience. However, it is worth pointing out that processing time is lower than network latency, so the minimization of the latter should be considered (for instance, by using the latest wireless communication technologies, like 5G) in AR environments where latency is critical [65].

## 6. Conclusions

This paper presented the design, implementation and evaluation of an open-source collaborative framework to develop teaching, training, and monitoring pediatric healthcare applications. The framework enables functionalities for connecting with other AR devices and for enabling real-time visualization and simultaneous interaction with virtual objects.

In order to show and asses the technical capabilities and the performance of the proposed open-source collaborative framework, an AR application was developed in order to demonstrate its potential for future researchers and developers of pediatric healthcare applications. Such an AR application actually consists of two applications: a mobile gaming application and a web application aimed at monitoring the progress of pediatric patients in terms of mood and performed activities. The collected data are shown in a user-friendly way through charts, so their representation is intuitive and easy to understand.

The developed AR system was evaluated by using two different mobile devices (e.g., a tablet and a smartphone) with different hardware capabilities. The conducted performance tests measured the latency and the processing time of every packet during real games. The obtained results show that average latency is always below 200 ms for every tested device, which results in a smooth gaming experience. However, it was also observed that the selected AR device impacts user experience substantially. In addition, it was concluded that wireless communications should be carefully examined in AR environments where network latency is critical. Nonetheless, considering the previous observations, it can be stated that the proposed open-source AR collaborative framework can help future researchers to develop the next generation of AR collaborative pediatric healthcare applications. As future work, the authors plan to conduct a thorough evaluation of the mobile app with pediatric patients as soon as the current COVID-19 pandemic situation allows it.

## Figures and Tables

**Figure 1 sensors-21-01865-f001:**
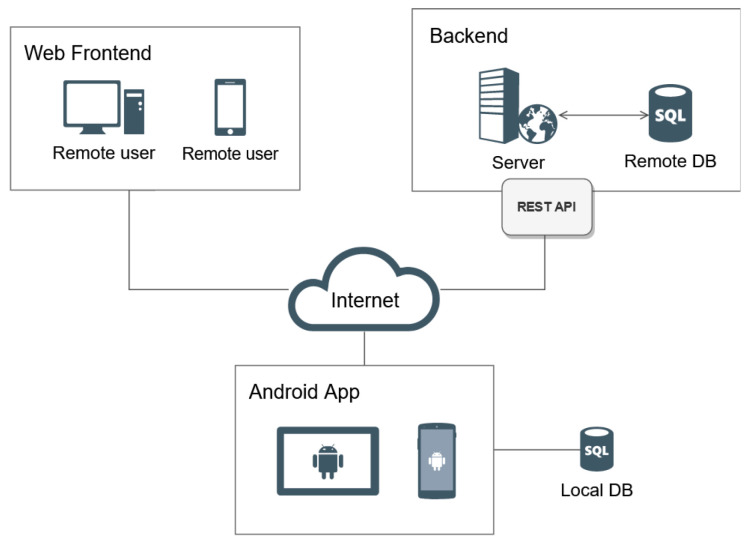
Overview of the system architecture.

**Figure 2 sensors-21-01865-f002:**
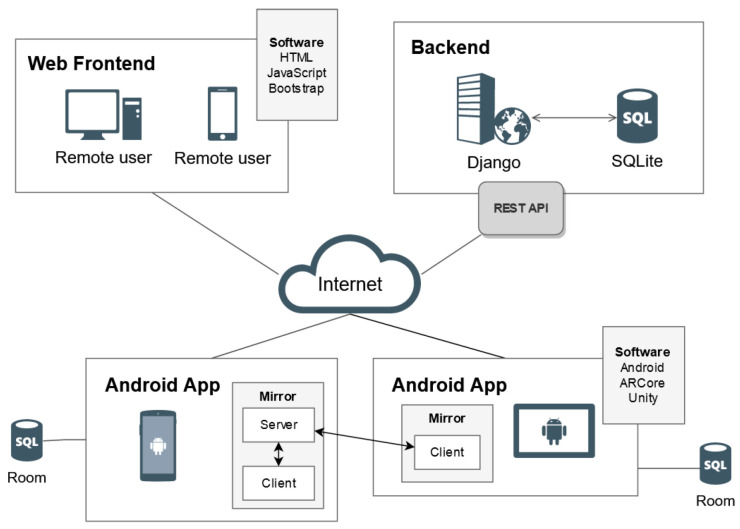
Implemented communications architecture.

**Figure 3 sensors-21-01865-f003:**
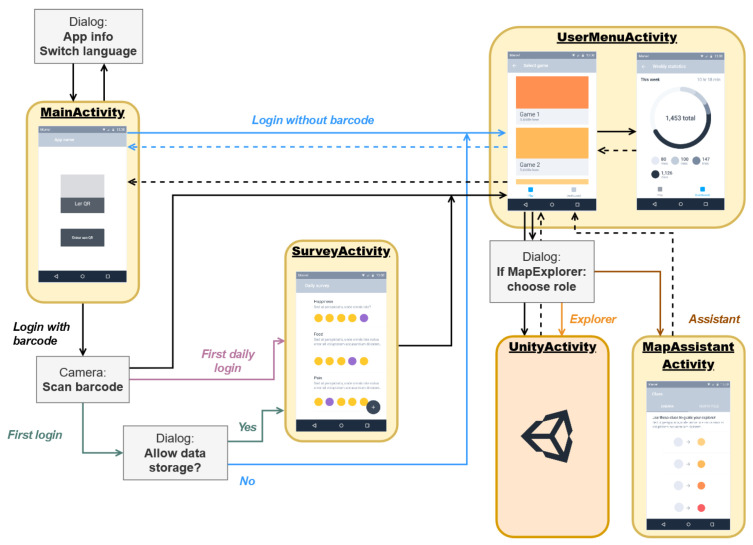
Overview of the mobile application components.

**Figure 4 sensors-21-01865-f004:**
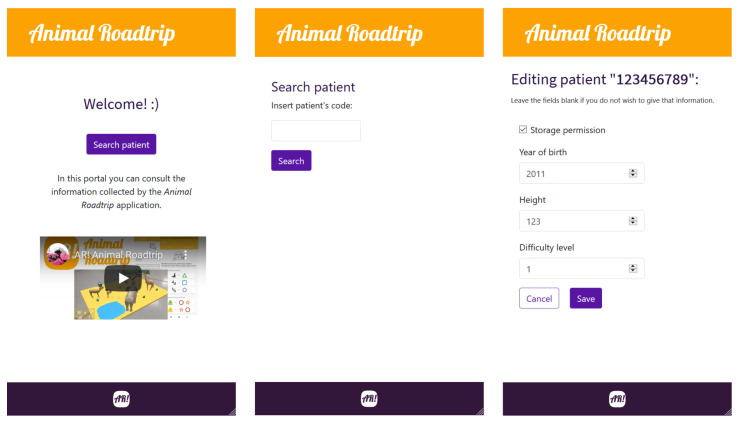
Overview of the Home page (**left**), Search page (**center**) and Edition page (**right**).

**Figure 5 sensors-21-01865-f005:**
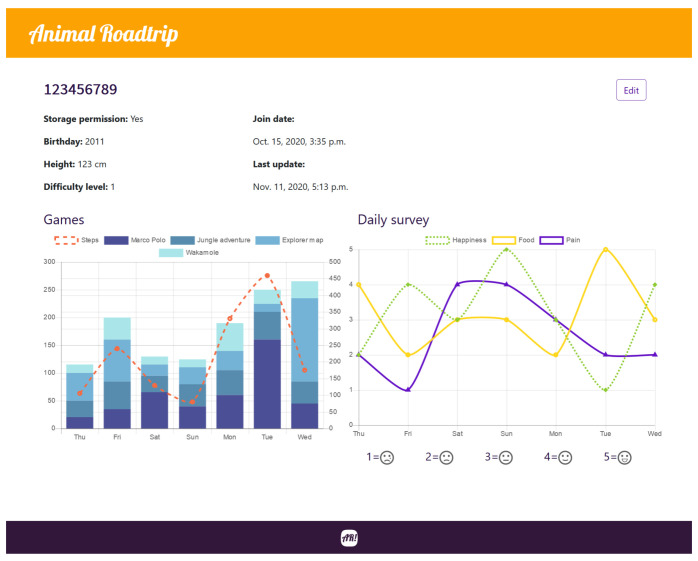
Patient data page of the developed web application.

**Figure 6 sensors-21-01865-f006:**
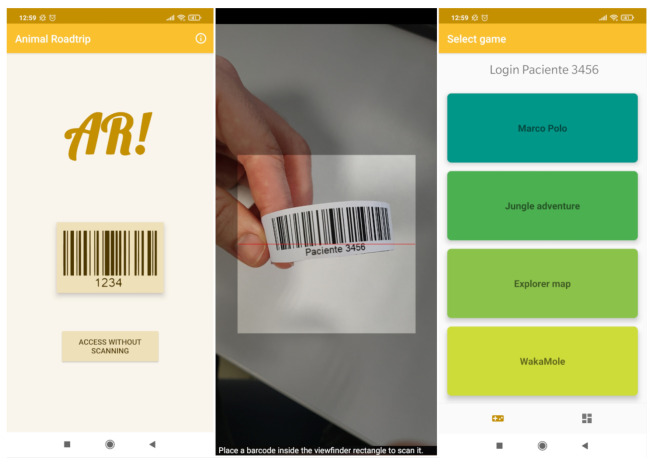
Overview of the login process.

**Figure 7 sensors-21-01865-f007:**
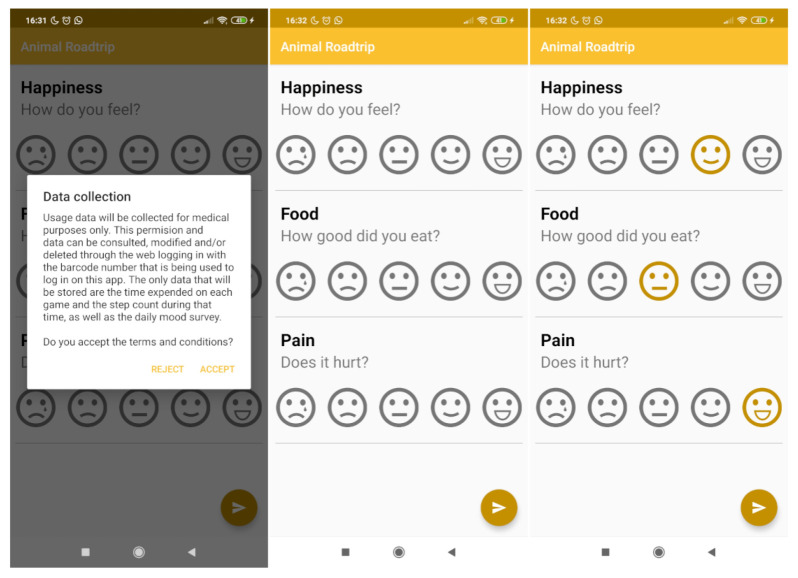
Overview of the daily survey process.

**Figure 8 sensors-21-01865-f008:**
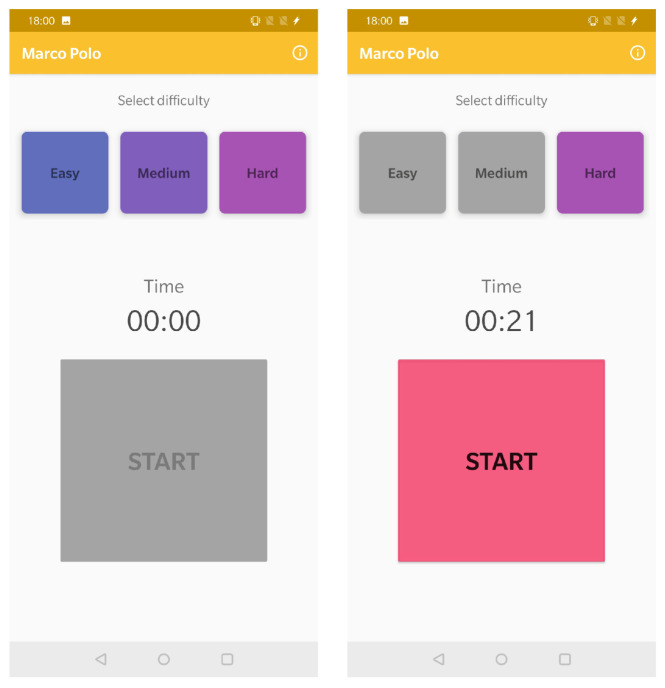
Overview of the “Marco Polo” activity.

**Figure 9 sensors-21-01865-f009:**
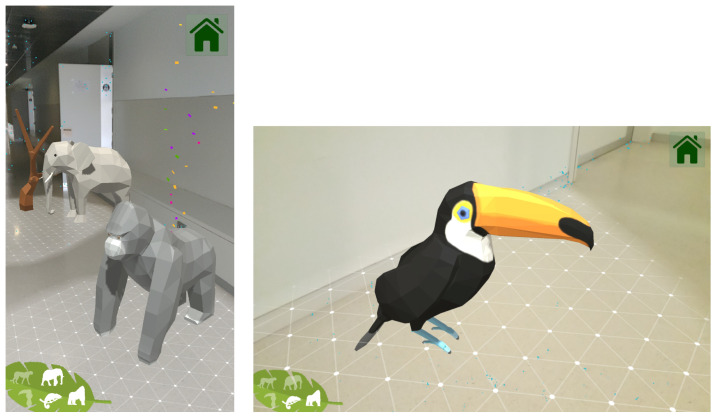
Screenshots from the “Jungle adventure” game.

**Figure 10 sensors-21-01865-f010:**
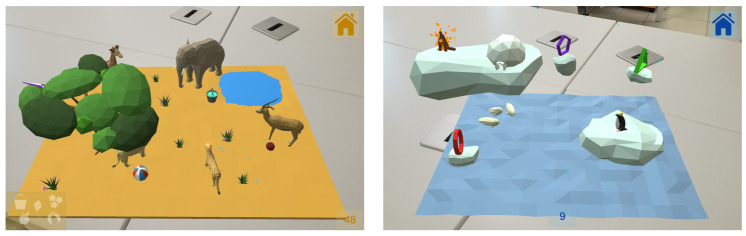
Screenshot from the “Savannah ” and “North Pole” maps.

**Figure 11 sensors-21-01865-f011:**
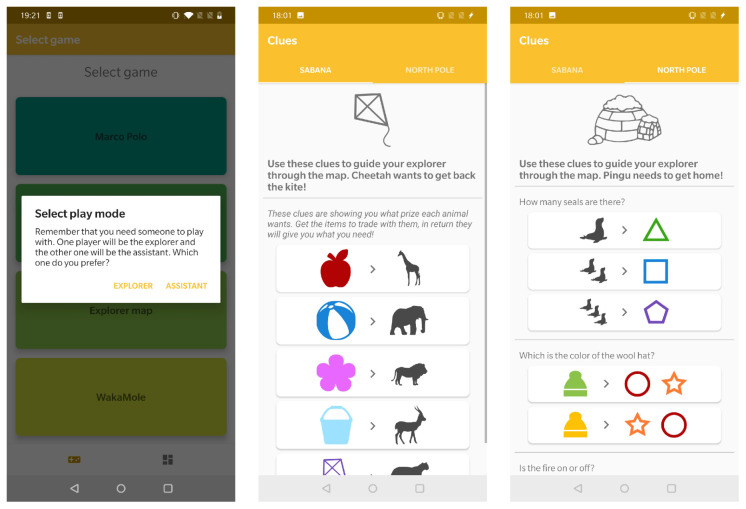
Overview of the “MapAssistant” activity.

**Figure 12 sensors-21-01865-f012:**
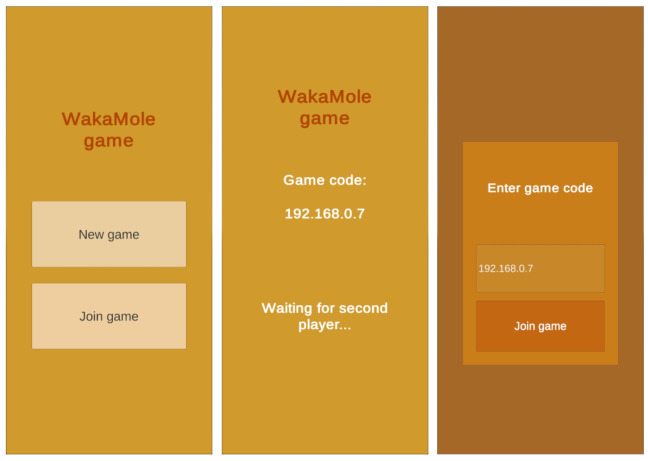
Overview of the Wakamole game matchmaking process: main menu (**left**), host screen (**center**) and client screen (**right**).

**Figure 13 sensors-21-01865-f013:**
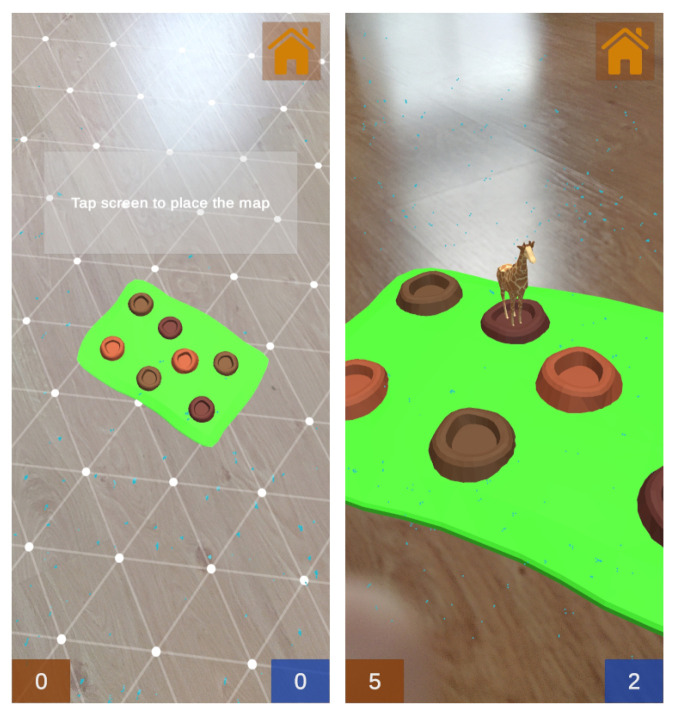
Screenshots from the Wakamole game: surface detection process (**left**) and during the game (**right**).

**Figure 14 sensors-21-01865-f014:**
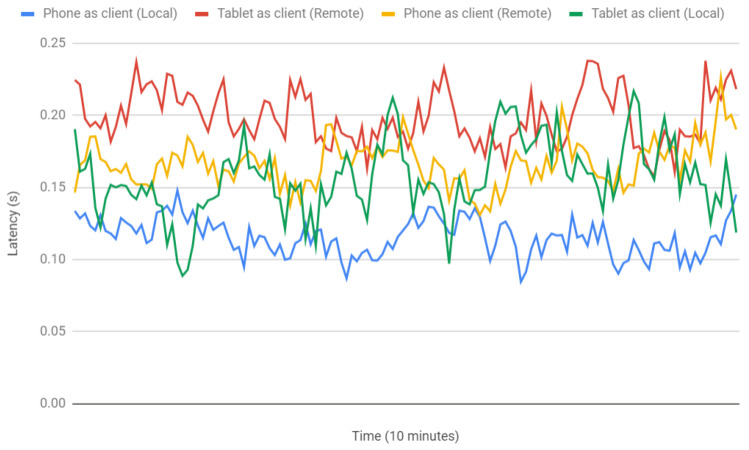
Latency comparison for two different devices on a remote and a local environment.

**Table 1 sensors-21-01865-t001:** Latency of the devices on each scenario (seconds).

Scenario	Device	Mean	Standard Deviation	Variance
Remote	Tablet	0.198936251	0.01881597504	0.0003540409168
	Phone	0.1667542366	0.01581829768	0.0002502185416
Local	Tablet	0.1568864479	0.02566691616	0.0006587905851
	Phone	0.1155596442	0.0127895142	0.0001635716735

**Table 2 sensors-21-01865-t002:** Processing time for client devices (seconds).

	Tablet	Phone
Mean	0.008257117669	0.003265850377
Standard Deviation	0.008145376438	0.003267552315
Variance	0.008138627362	0.003264794

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
