# Peer review of "Developing the Next Generation of Augmented Reality Games for Pediatric Healthcare: An Open-Source Collaborative Framework Based on ARCore for Implementing Teaching, Training and Monitoring Applications"

_sensors, 2021, doi:10.3390/s21051865_

Round 1

Reviewer 1 Report

The paper presents a complete framework to design augmented reality "exergames" for hospitalized children. The paper's main contribution is the development of a complete framework by taking into account both the AR game and all the steps necessary to have a working system. I have appreciated the effort towards having a good user interface, not relying on Unity, only and the communication part.

The authors also discussed the limitation (e.g., for what concerns the wireless connection).

The authors also commented on the fact that they did not perform user studies with final users. This is not the scope of the paper, so I do not believe it is a problem.

The only thing is missing is a discussion and an analysis of the limitation of ARCore in several environmental and working condition. Are the games working in a wide range of conditions (illumination, textures on the floor or the table, ...)  I would suggest discussing this point.

Author Response

Dear Sir/Madam,

The authors would like to thank the reviewer for his/her valuable comments, which have certainly helped us to improve the manuscript. Please find attached our detailed responses to the comments. In order to ease the labor of the reviewers we have colored in red the differences with the previous version of the article.

Regards,
The authors.

Reviewer 2 Report

Thé paper is presenting an AR based gamification healthcare application for paediatric patients.

the abstract and introduction are quite messy and hard to follow. Statements were made without providing the sources/references.

Background (existing, motivation, problem statement, etc. ), study design, study methodology,  results, discussion, and an adequate conclusion are missing. Though, the authors provide a good architecture and design of the system and conclude accordingly.

Some figure must be redrawn.  

The paper contient is rather an applications blueprint than a scientific report.

The authors are strongly advised to clearly provide the motivation and objectives as well as aims of their work. They are further advised to set the research methods, questions, and conduct a comparison with existing gamification app in paediatric. Correlation needs to be calculated.

Otherwise, they should remove completely the context and thus present the design and architecture of their system and add a section  like "Application Domain/field/area". In that section they could show how this application can benefit to paediatric patient.

Author Response

(The authors gave the same response as above.)

Reviewer 3 Report

The manuscript presents new AR games for pediatric healthcare. However, the authors did not conduct any performance evaluation using children. What is essential is not an evaluation in terms of latency and processing time but actual benefits. The reviewer agrees with the authors' view, but data showing actual benefits for children is needed. Thus, the reviewer thinks this paper is too early to be published.

Author Response

(The authors gave the same response as above.)

Reviewer 4 Report

The authors have presented a set of augmented reality games for pediatric healthcare. The importance of the work is clear, the paper is well written in a clear and detailed manner.

The main problem of this paper is described by the authors themselves: “that this article is focused on providing an extensive description of the proposed system and of its performance evaluation, so no empirical scientific  tests with children are provided for the validation of the system. Such results will be  presented and analyzed in a future work as soon as the COVID-19 pandemic allows for  carrying out the experiments in safe scenarios.”

Although all the paper is well written and presented, it has no novelty whatsoever; it is a very well and in-depth software description than an academic paper .

Moreover, all content and presentation was not compared with traditional approaches and the authors should, at least, theorize the benefits of such application. How does the usability will help decrease stress? How long this system can be used to optimize the user experience? Is this will help to detach even more the youth from reality? What are the expected side effects (there are always)?  how perception and environmental sensing of children with such needs were captured as a user’s requirements?

 I do understand that covid-19 had a huge impact over research all around the globe,  but an important research such yours should focus on its application and design, more than basic software documentation

Finally, in this context, the conclusion is dangerous. There are  lines 

“This article presented an autonomous AR system for children devoted to improve their mood and to encourage physical activity”

"for monitoring remotely the progress of the patients in terms of mood and performed activities"

"the developed system can provide support for doctors and families, who can monitor patients, extracting data in a non-invasive way by taking advantage of the fact that children are enjoying their time and doing physical activities at the same time"

that have no related result or proof, nether a theoretical one.

Therefore, despite all the qualities and the importance of the work, it should have another focus and, dealing with the desirable “clients” should much more and carefully analyzed, even in a theorical manner.

Author Response

(The authors gave the same response as above.)

Round 2

Reviewer 2 Report

I see the improvement. Though, there is still more room for improvement.

You are better to rewrite the paper and point out the novelty, which you do not really point out at this stage. You also need to choose between presenting a technical document and reporting a research work. You still mixing both together.

Anyway, the paper is acceptable in this form.

Be advised that you can split this paper in two: one technical part and publish it. The second needs to be supported by a proof of concept where you need to collect data, test your framework, compare the results with existing systems to point out the novelty and improvement your work provides.

Author Response

Dear Sir/Madam,

The authors would like to thank the reviewer for his/her valuable comments, which have certainly helped us to improve the manuscript. Please find attached our detailed responses to the comments. In order to ease the labor of the reviewers we have colored in red the differences with the previous version of the article.

Best regards,
The authors.

Reviewer 3 Report

I am sorry to say that, but the authors did not answer my previous comments in the revised manuscript.

Author Response

(The authors gave the same response as above.)

Reviewer 4 Report

The authors have improved the paper. Although it is not at the level that the research deserves (again, it is a significant paper), this reviewer understands the impact that covid-19 has over practical results. Finally, I understand that this paper is more related to divulge the preliminary work than the final research itself.

As a minor suggestion, I would ask to add more comments over how you will validate the software. The procedure, tests, how many pacients, for how long, etc...

Congratulations on the work.

Author Response

(The authors gave the same response as above.)
